# Position: AI Agents Need Authenticated Delegation

**Tobin South** [1] **Samuele Marro** [2] **Thomas Hardjono** [1] **Robert Mahari** [1 3] **Cedric Deslandes Whitney** [4]
**Alan Chan** [5] **Alex Pentland** [1 6]

## Abstract

The rapid deployment of autonomous AI agents creates urgent challenges in the areas of authorization, accountability, and access control in task delegation. This position paper argues that authenticated and auditable delegation of authority to AI agents is a critical component of mitigating practical risks and unlocking the value of agents. To support this argument, we examine how existing web authentication and authorization protocols, as well as natural language interfaces to common access control mechanisms, can be extended to enable secure authenticated delegation of authority to AI agents. By extending OAuth 2.0 and OpenID Connect with agent-specific credentials and using transparent translation of natural language permissions into robust scoping rules across diverse interaction modalities, we outline how authenticated delegation can be achieved to enable clear chains of accountability while maintaining compatibility with established authentication and web infrastructure for immediate compatibility. This work contributes to ensuring that agentic AI systems perform only appropriate actions. It argues for prioritizing delegation infrastructure as a key component of AI agent governance and provides a roadmap for achieving this.

## 1. Introduction

**This position paper argues that when AI agents interact with third parties—such as digital services, other AI agents, or humans—those parties must be able to verify both the principal delegating authority to the agent and the precise scope of that delegation.**

Agentic AI systems (or 'agents') are useful for their ability to interact with other actors on a user's behalf and accomplish complex tasks autonomously, including by interacting with a variety of external digital tools and services (Nakano et al., 2021; Lieberman, 1997; Fourney et al., 2024). For example, AI agents prompted to book travel arrangements for a holiday may browse the web for recommendations, search for flights via APIs, or message an airline agent in natural language via chat services to arrange a booking. Such communications could even extend to AI agent negotiations (Abdelnabi et al., 2023) and other multi-agent contexts.

Many risks exist for AI agents, including challenges from prompt injection attacks (Perez & Ribeiro, 2022; Liu et al., 2023), risks to contextual confidence (Jain et al., 2023), risks from not communicating properties of AI agents to third parties (Chan et al., 2024a), unreliability of distinguishing humans online (Adler et al., 2024), challenges with human in the loop (European Commission, 2021; Gabriel et al., 2024), and other challenges arising from incomplete governance and transparency (Shavit et al., 2023; Reuel et al., 2024; South et al., 2023). To address these, the world needs ways to explicitly delegate authority to agents, transparently identify those agents as AI, and enforce human-centered choices around security and permission for these agents.

This work has three key contributions. First, section 2 builds upon the existing literature to argue **why authenticated delegation is important** for AI agents and what risks it could mitigate. Second, section 3 directly addresses this need, outlining how to **extend existing authentication and authorization protocols to enable authenticated delegation** for AI agents, examining the role OpenID Connect and OAuth 2.0 could play in enabling a pragmatic, robust, and extensible implementation. Third, section 4 explores the **role of agentic access control** and outlines a method for **expressing flexible, natural language permissions for agents** and transforming them into auditable, fine-grained access control rules, that can operate across agent modalities (e.g., web requests, computer use, or language interfaces). Further, this work provides **example use cases** of the framework in Appendix E and a **legal analysis of the implications** of this work in Subsection 5.3.

---

[1]MIT, Cambridge, MA, USA [2]Department of Engineering Science, University of Oxford, UK [3]Harvard Law School, Cambridge, MA, USA [4]University of California, Berkeley Berkeley, California, USA [5]Centre for the Governance of AI, Oxford, UK [6]Stanford University, Palo Alto, CA, USA. Correspondence to: Tobin South <tsouth@mit.edu>.

*Proceedings of the 42$^{nd}$ International Conference on Machine Learning*, Vancouver, Canada. PMLR 267, 2025. Copyright 2025 by the author(s).

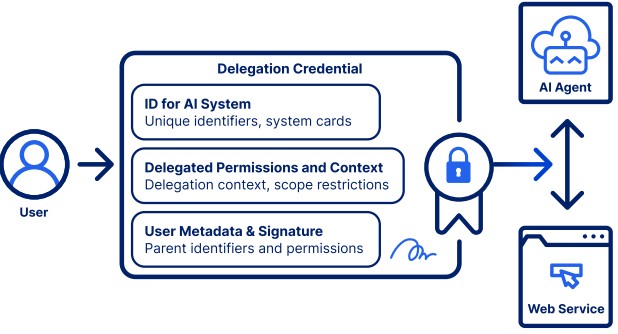

*Figure 1.* Conceptual overview of a verifiable delegation credential for AI agents. Users issue delegation credentials that include: the AI system's unique identity and properties, delegated permissions with contextual scope restrictions, user metadata, and cryptographic signatures for verifiability. These credentials enable secure, trustworthy interactions between AI agents and third-party services, ensuring traceability and appropriate delegation of authority.

## 2. Why authenticated delegation is important

Authenticated delegation is the process of instructing an AI system to perform a task that requires access to tools, the web, or computer environments in such a way that third parties can verify that **(a)** *the interacting entity is an AI agent*, **(b)** *that the AI agent is acting on behalf of a specific human user*, and **(c)** *that the AI agent has been granted the necessary permissions to perform specific actions.*

We distinguish three key concepts: *authentication* confirms an entity's identity; *authorization* determines the permissible actions and resource accesses that the authenticated identity is allowed to perform, defining the scope and limitations of delegated activity; and *auditability* allows all parties to inspect and verify that claims, credentials, and attributes remain unaltered, supporting trustworthy authentication and authorization decisions.

Verifying the properties of interacting entities will be relevant whenever a context exists where an AI agent *could* act on behalf of a human user, especially where the agent can take consequential actions. This remains true whether the AI system is run locally or provided by an AI vendor—as harm can occur in both—and must be able to operate across various digital contexts and with AI models of heterogeneous capabilities.

At a high level, authenticated delegation involves a human user creating a digital authorization that a specific AI agent can use to access a digital service (or interact with another AI agent) on behalf of the user, which can be verified by the corresponding service or agent for its authenticity. Such authorization can include additional information, such as unique identifiers for the agent instance, permissions on

what the agent is allowed to do, and other information (e.g., the capabilities and failure modes of the agent or information about the human user).

In practice, this approach does not need to be substantially different from existing authentication and authorization mechanisms used today, such as how a calendar application is authorized to access a user's calendar data and scan it for upcoming events. However, AI agents' autonomous and highly capable nature means more care is needed in how we manage delegation. As such, let us examine the use cases for authenticated delegation in more detail.

### 2.1. Arguments for authenticated delegation

Authenticated delegation opens avenues for AI agents to accelerate complex tasks, automate workflows, and seamlessly interface with digital services on behalf of human users. However, granting such agency also entails risks around scope misalignment, resource abuse, or a breakdown of clear accountability.

**Securing tool use and web access**   A key aspect of AI agent deployment is the ability to use tools or access external services. For simple tasks such as asking an agent to search the web for information, write and execute code, or generate an image, this is straightforward and does not require additional authorization or individual-specific security mechanisms. However, to unlock use cases such as interacting with personal or organizational accounts, accessing sensitive personal information, or interacting with consequential infrastructure, more robust delegation frameworks are needed.

**Example:** Consider an AI agent booking a holiday. Searching the web for information may not need authorization, but how could that agent access a user's calendar or make a purchase? For calendars, users are used to the expected flow of granting access to applications to access their calendar data. This would be no different for an AI agent (and would be naively supported in the solution outlined in section 3)– indeed, limited OAuth 2.0 support is enabled in some agent tools such as OpenAI GPT actions. Now consider a flight purchase. You *could* provide your credit card details in the context window for the agent and prompt it to follow the budget, but this introduces a variety of security concerns and is dependent on the underlying reliability of the AI system to not take unexpected actions or be vulnerable to attacks or jailbreaks. Instead, an AI agent should be authenticated and authorized to make a purchase on specific booking services, where credit cards are stored securely, and where explicit spending limits can be enforced.

**Communicating limitations and restricting scope**   Current approaches to limiting the scope of AI agents are limited

and one-sided. A user can provide a strong prompt to an agent to limit its actions, but this comes with a variety of failure modes (Liu et al., 2023). Access to tools or websites can be blocked, but the granularity of these control systems is limited. An AI system deployer could implement further controls, such as monitoring and blocking specific actions or website subdomains when agentic functionality occurs, but doesn't communicate these limitations to the service the agent is interacting with. A more detailed examination of how this could be designed across web, API, and natural language access modalities is available in section 4.

**Example:** An AI agent is used by a physician to provide diagnostic recommendations in a telemedicine portal, logging in with basic credentials that do not specify its limitations. The portal assumes full physician capabilities, granting the agent access to all patient records, including a video with a voice recording from a specialist. The agent, being text-only and unable to process video, generates a diagnosis based solely on the text data. Trusting the incomplete recommendation, the physician risks making a misinformed treatment decision. If the agent's limitations were explicitly communicated via authenticated delegation, the portal could have flagged the need for a human review of the multimedia content, avoiding a potentially harmful oversight.

**Verification in multi-agent communication** When AI agents communicate to collaborate on tasks or facilitate interactions, ensuring mutual authentication becomes paramount. Securing communication channels is not enough; agents must also verify that they authentically represent the users or organizations they claim to represent. Mutual authentication ensures that agents can trust each other's intentions, capabilities, and authority, which prevents impersonation, unauthorized actions, and potential misuse.

**Example:** Two AI agents—one representing a hospital and the other an insurance company—collaborate to process a patient's claim. Without mutual authentication, a third-party malicious agent could impersonate the hospital, submitting fraudulent claims, or the insurance agent could reject valid claims out of concern over authenticity.

**Protecting human spaces online** As AI agents grow increasingly adept at mimicking human behavior—crafting text, creating personas, and even replicating nuanced human interactions—it becomes harder to maintain digital environments genuinely inhabited by real people. This challenge drives the need for safe, human-only online spaces where authenticity is preserved (Adler et al., 2024). However, many AI agents act as useful proxies, assistants, or representatives for human users who cannot, or prefer not to, engage directly. Authenticated delegation enables these spaces to be selectively accessible to AI agents, while still ensuring

that the AI agents are linked to verified human principals. This tool is also more granular than simple bot detection, user-agent restrictions, or `robots.txt` limitations.

**Example:** Some websites wish to block 'bots' or restrict access to specific uses (based on age, nationality, etc). By design, any such restriction will also block AI agents. Instead, platforms could explicitly allow AI agents to access their services in controlled ways by leveraging authenticated delegation. This approach would ensure that AI agents act transparently on behalf of verified human users. For instance, an agent could access a user's social media account to retrieve information about friends and help draft an email, all while maintaining compliance with platform policies and ensuring accountability.

**Supporting contextual integrity** Contextual integrity addresses adherence to context-specific norms and privacy, which include actors (who is involved in the information flow), attributes (what information is shared), transmission principles (under what conditions information is shared), and social context (the broader cultural, institutional, or situational environment shaping these norms) (Ghalebikesabi et al., 2024; Zhan et al., 2022; Fan et al., 2024; Nissenbaum, 2004). Contextual integrity offers a perspective for reasoning about how AI agents can act in ways that are contextually appropriate, transparent, and aligned with societal norms and the expectations of their human delegators (Bagdasarian et al., 2024; Ghalebikesabi et al., 2024; Bloom & Emery, 2022). This includes exploring which decisions can reasonably be made autonomously by the AI and under what conditions human oversight or intervention might be necessary (e.g., when is human-in-the-loop required and who is responsible).

**Example:** An AI assistant with authenticated delegation can be issued distinct credentials for separate contexts (e.g., an enterprise-context assistant and a personal one). By enforcing these scoped credentials, services can ensure that the assistant adheres to contextual integrity and rejects actions that cross boundaries, such as using information from work documents to complete personal forms.

## 2.2. Background

Authenticated delegation can address various challenges, from the traceability of AI outcomes to limitations on what spaces can be accessed and actions taken by AI systems. The overarching aim of identification and credentialing systems is to facilitate secure online environments and authenticated access to services. To this end, various protocols and standards have been developed, tailored to both human users and AI systems, to uphold these goals in different contexts.

Subsection A.1 outlines existing literature and tools for identifying AI, their content, and other humans users (e.g.,

tools for verifying human identity online, tools for proving personhood, and tools for tracking AI system outputs, watermarking, frontier AI access control, and AI identifiers).

In addition, Subsection A.2 outlines existing literature and tools for documenting AI systems and the data that create them, a useful precursor to identification and credentialing for AI agents.

For a discussion of the governance of AI agents and dangerous capability management (an issue we do not address here but could benefit from authentication and authorization), see Subsection A.3.

**Comparisons to Model Context Protocol and GPT Operator**   One example of an AI-centric protocol is the Model Context Protocol (MCP) (Anthropic, 2024) from Anthropic, which enables secure, structured interactions between AI systems and external tools or data sources. MCP aims to enhance the contextual relevance of AI outputs by establishing a standardized framework for connecting models to resources to facilitate applications like retrieving live data, interacting with APIs, and executing tasks in real-time.

While a useful standard, it does not fully address the needs of authorized delegation, enabling only system communication and basic access controls rather than broader authentication and identity management. LangChain's Agent Protocol / LangGraph Platform extends this idea to enable multi-agent interoperability.

OpenAI's Operator takes a different approach, operating a web browser to interact with the web. This allows users to log in but stores login credentials as cookies in the browser, an insecure approach that limits the user's ability to revoke access, control permissions, and audit the actions of the operator post-hoc.

**Prototype Implementations and Emerging Standards:** Recently–including in response to early pre-prints of this work–a blossoming ecosystem of prototype implementations has emerged, exploring authentication and authorization tools for AI agents. Many of these efforts are centered around the Model Context Protocol (MCP) from Anthropic or Agent-to-Agent Authentication (A2A) from Google. These prototypes primarily focus on authentication and basic scoped authorization, with limited exploration of robust agent identifiers. Furthermore, they often rely on standard one-time browser-based approval flows, which may not scale effectively with the increasing number of agent-tool interactions.

**How authenticated delegation combines these solutions** This argument suggests that authenticated delegation combines and extends existing approaches—AI agent IDs and credentials, proof-of-personhood and identity verification

for human users, and content provenance and watermarking methods—to form a cohesive framework. This approach inherits well-established practices for identity management while introducing explicit scoping and metadata for AI agents. This integration allows for granular, enforceable permission sets, clearer accountability chains, and richer context signals (like a model's certifications or limitations) to be attached to each delegated action, with a more robustly verifiable construction than a simple agent ID system card. In effect, authenticated delegation complements existing standards and enhances their reliability by anchoring the actions of AI agents to verifiable human principals and recognized AI-specific credentials, creating a unified foundation for safe and accountable AI interactions. To this end, section 3 introduces a concrete framework with additional security guarantees to package these elements together in a robustly verifiable way.

# 3. Extending OpenID Connect for identifying and authenticating AI agents

To support the argument of section 2, this section proposes a concrete technical framework building on existing internet-scale authentication protocols to introduce mechanisms for delegating authority from users to AI agents and describes a token-based authentication framework that leverages OpenID Connect and OAuth 2.0.

## 3.1. OAuth2.0 and OpenID-Connect

While new frameworks for AI system identification are emerging, there are valuable lessons to be learned from existing internet-scale authorization and authentication protocols. In particular, the OAuth 2.0 protocol (Hardt, 2012) and its extensions provide battle-tested patterns for delegated authorization and identity verification that could inform the development of AI agent credential systems.

OAuth 2.0 emerged from the need for users to provide authorization to one service to access resources located in another service, based on the RESTful paradigm (Fielding, 2000). A key requirement underlying OAuth 2.0 is the ability for access to be continually granted even if later the user is absent (e.g., offline).

The wide deployment and popularity of the OAuth 2.0 protocol enabled new features and extensions to be added. One successful extension—namely the *OpenID-Connect* protocol (OIDC) (Sakimura et al., 2014)—is the addition of flows dealing with the user authentication. The service dealing with authentication is referred to as the *OpenID Provider* (OP). A key addition introduced by OpenID-Connect is the *ID-token*, which carries information about the human user that can be retrieved from the OP (i.e., by presenting ID-token). Here a merchant (as the Relying Party) would input

the ID-token to the relevant token-validation endpoint at the OP in order to obtain more information about the user. We believe this capability may be extended to address the case of AI agents.

Another extension of the OAuth 2.0 protocol that enables a user to manage multiple resources distributed across many Resource Servers is the *User-Managed Access* (UMA) protocol (Hardjono et al., 2015). The UMA model may fit use cases where the human user possesses multiple AI Agents and where a single point of policy or rule configuration is desirable (Hardjono, 2019). Here, the AI Agents can be viewed as distributed resource servers owned by the user. Using the UMA Authorization Server, the user can set policy at one location and automatically propagate these policies to the multiplicity of AI Agents.

### 3.2. Delegation of authority from the user to the AI agent

Given that the OAuth 2.0 protocol is an authorization protocol, it is worthwhile considering reusing the OAuth 2.0 patterns to establish a new mechanism for the human user to *delegate* specific tasks to the AI Agent. In other words, the human user authorizes the AI Agent to carry out certain limited-scope tasks on behalf of the user.

In this proposed extension, the human user must first authenticates with the OpenID Provider (OP) to demonstrate their identity. The user then 'registers' the AI Agent to the OP so that external entities who later seek to obtain further information about the AI Agent can do so to the OP. Registration could be done automatically in the background when an agent is created through a vendor (such as creating a new assistant instance with OpenAI).

Existing OAuth 2.0 client registration protocols can be customized to enable the user to register the AI Agent to the OpenID Provider and designate the AI Agent as a delegate or surrogate of the human user.

Next, the human user can issue a new *delegation token* that authorizes the AI Agent to carry out tasks on behalf of the user. Here, the term 'authorize' is utilized to explicitly call out the fact that the AI Agent is owned (driven) by a human delegator. For details on what could be included in the delegation and agent ID tokens, see Subsection B.1.

Both the user ID-token and the AI Agent delegation token can be referenced from within (or even copied into) a W3C Verified Credentials (VC) data structure (Sporny et al., 2022). This enables the AI Agent to wield the VC in its interactions with other entities (e.g., other services or other AI Agents), and have the benefit that both tokens would be verifiable at the standard OP.

It is worth noting that these delegation and authentication ex-

changes could alternatively be implemented using W3C VC issuance and delegation mechanisms. In such a scenario, a W3C VC could generate an OpenID-compatible credential, enabling seamless interfacing with OpenID systems. While this integration highlights the interoperability between W3C VC and OpenID ecosystems, further exploration and formalization of this process are beyond the scope of this paper and left as future work. See Subsection B.2 for more details.

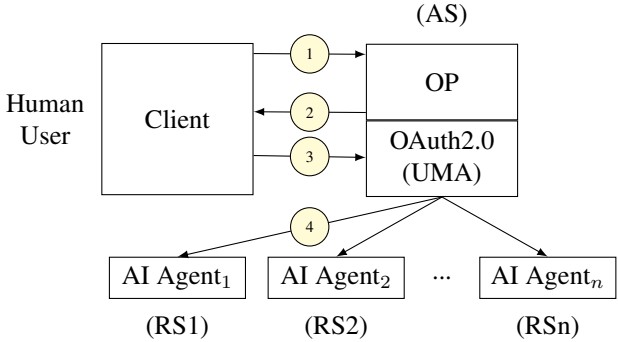

*Figure 2.* Integration of OpenID Connect (OIDC) and User-Managed Access (UMA) protocols for establishing delegated authority from human users to AI Agents. The diagram illustrates the authentication flow where a human user first authenticates to an OpenID Provider (OP) (1 & 2), registers their AI Agent (3), and issues a delegation token (4). This token empowers the AI Agent to perform authorized tasks on behalf of the user. The verification of both the user's ID token and the AI Agent's delegation token can be performed through the standard OpenID Provider, leveraging existing OAuth 2.0 patterns while incorporating new delegation mechanisms for AI Agent authorization.

## 4. Defining scope and permissions for AI agents

Authenticated delegation is inherently tied to robust scoping mechanisms. Users must be able to specify their permissions and instructions clearly and unambiguously. This directly conflicts with the extremely large possible action space AI agents can perform.

While much work in reliability and alignment focuses on ensuring that AI agents follow instructions correctly, the risks of misinstruction, prompt injection attacks, and reduced security auditability make pure natural language prompts an incomplete scoping, permission, and security tool.

### 4.1. Combining structured permissions, natural language, and user oversight

**Resource scoping as a foundation.** We argue that the most broadly applicable strategy for access management connected to authenticated delegation is to enforce *resource*

*scoping with structured permissions.* The brittleness of natural language mechanisms makes them unsuitable for production-level usage of AI agents, especially when security or compliance is a concern. In contrast, structured permissions are unambiguous and deterministic, providing verifiable guarantees against unauthorized access. Focusing on resource scoping also significantly reduces the overhead of specifying every authorized task in detail. To an extent, agents could attempt to represent task-scoping instructions in the form of resource scoping, using domain knowledge of the contexts in which they operate. Since resources are generally discrete and can be classified, enumerated, and grouped into domains, controlling resource access implicitly prevents many potential tasks that would require out-of-scope resources. Additionally, structured resource scoping has several advantages:

- It does not depend on how a user delegates tasks—be it via a script, an AI agent, or a more traditional workflow;
- It is compatible with existing non-AI access control systems, which focus on machine-readable permissions for resources (e.g., databases or URLs);
- It is suitable for structured logging and version control, which simplifies auditing and compliance reporting.

Though users may supplement resource scoping task constraints written in natural language, the core resource-based policies provide a safety net that is largely immune to ambiguities in language or model vulnerabilities. In other words, even if an LLM or another AI agent is tricked or misaligned, its ability to execute harmful actions is constrained by the underlying resource permissions.

**Connecting to natural language.** While robust and auditable, structured resource scoping alone lacks ease of use and flexibility. To address this, the instructions for the LLM (or a separate scoping prompt) can flexibly express the scoping limitations that should be applied. These natural language scopes can be converted to a structured scoping format by the agent or an AI system in the corresponding environment (which has more detailed knowledge of the relevant resource profiles). Examples of conversion between natural language and structured permissions include Subramaniam & Krishnan (2024), which generates PostgreSQL restrictions, and Jayasundara et al. (2024), which uses retrieval to generate custom JSON policies.

A similar process could also be performed for different environments and digital services an agent interacts with, allowing a flexible set of permission instructions to be applied across a wide range of services and contexts (which is important given the broad action space of AI agents).

**Bringing a human in the loop.** The key final step is validating these structured access controls via the human delegator. Authorization workflows present an opportunity for users to briefly review and approve structured access control limitations for different systems. For instance, in Wright (2024) LLM agents agree on structured information (in this case, meeting dates) which are then confirmed by human users.

**Combining into a hybrid implementation.** Bringing these elements together into an implementation is relatively straightforward. An LLM assists in converting high-level, natural language resource constraints into formal, structured rules that users can subsequently review and approve. **For example:**

1. A user writes: "Allow the agent to read and write to the directories about 'projectAlpha', but do not grant it access to the folders with financial folders;"
2. The LLM translates this requirement into a policy definition, either in a universal permission language (e.g., XACML) or in the specific permission language used by the resource (e.g., SQL access policies for databases). In this specific case, the LLM enumerates "projectAlpha" resources while explicitly denying access to "financials2023;"
3. The user reviews, corrects if necessary, and finalizes the policy.

While many specific details of such a workflow need to be addressed, such as intermediate validation checks and the evaluation of the robustness of LLM translation into structured languages, we leave these specifics to future work.

Focusing on structured, unambiguous resource constraints is the most reliable way to ensure that an AI agent remains within authorized bounds in a given environment. While there is still room for higher-level (often natural language) task constraints, these should be treated as guidance towards the primary enforcement mechanism. Indeed, while natural language can adequately address the extremely large possible space of agent actions, its transformation into access controls grounds the limitations on agent actions into finite auditable controls. Structured resource scoping reduces the reliance on model alignment alone, decreases the risk of adversarial prompt injections, and simplifies the integration with well-established security mechanisms. Combining this approach with well-designed authentication flows and helping the user interpret the generated policies can reduce the chances of human errors, enhance accountability, and improve the robustness of authenticated delegation.

Appendix C provides a more detailed discussion of extending scoping, outlining the background of structured permission languages and authentication flow dynamics and explaining how natural language can be mapped to task and

resource scoping across modalities.

# 5. Discussion

## 5.1. Limitations in the technical proposal

The technical proposal builds on existing technologies to address the unique challenges of AI agent delegation but comes with a range of limitations.

As detailed in Subsection D.1, relying on OIDC introduces repeated sign-in overhead, centralizes privacy risk with providers, and can be overly complex when simpler alternatives like W3C Verifiable Credentials or GNAP exist.

Meanwhile, as described in Subsection D.2, natural language scoping risks unreliable policy translations, opens new LLM-based attack surfaces, suffers from contextual drift, and in some cases relies on third parties for correct enforcement.

## 5.2. Can model vendors provide this?

Model vendors (e.g., OpenAI, Anthropic, Google) can provide tooling to share which user is being represented when an AI system accesses a digital service and the intended scope or permissions. This is encouraged. However, current approaches to sharing such information are insufficient from a security and verifiability perspective, such as including the information in the user-agent string of the AI system or writing the information into API calls made by the AI system. Instead, these services could act as an OpenID Provider (or partner with one) for the AI system without any change to the user experience; alternatively, if they prefer a different instantiation of the authenticated delegation framework, they could provide W3C verifiable credentials paired with robust, unique IDs for AI agents and users.

Implementing authenticated delegation is also feasible when AI systems and agents are self-hosted or deployed on custom infrastructure. This includes leveraging internal identity management infrastructure for human users and incorporating custom permission controls. Such systems can operate internally within an organization to ensure AI system usage aligns with identity and access management (IAM) policies and delegation frameworks across various technology stacks and modalities.

## 5.3. Legal grounding for authenticated delegation

The law of agency addresses circumstances in which one party, the principal, authorizes another party, the (human) agent, to act on their behalf (Garner, 2019). At its core, agency law determines when a principal may be held liable for the acts of their agent, ensuring that third parties are not unfairly disadvantaged by having to ascertain who holds ultimate responsibility.

A key result of agency law is to instill trust and confidence in market transactions: by providing clear rules about liability and authority, agency law reduces uncertainty and contributes to more efficient market operations (Posner, 2019; Williamson, 1975; Casadesus-Masanell & Spulber, 2005).

One central concept in agency law is that of "apparent authority," extensively discussed in the Restatement (Third) of Agency (American Law Institute, 2006). Under this doctrine, a principal can be held responsible for acts that a reasonable third party perceives the agent to be authorized to perform, even if the principal never granted that authority explicitly. This principle also helps maintain market stability: third parties need not investigate every aspect of an agent's credentials or verify each claim of authority before proceeding with a transaction as long as the agent appears to be acting on behalf of the principal in a reasonable manner.

It remains uncertain how established agency doctrines will adapt to AI agents that can learn, self-modify, or operate autonomously (Balkin, 2015; Adler et al., 2024). Traditional notions of intent, consent, and observable authority are difficult to apply to current autonomous systems. In response to these uncertainties, the authenticated delegation framework offers a model in which each delegation of authority is verifiable. Rather than relying on appearances, this framework enables third parties to automatically confirm that an AI agent is indeed authorized to act on behalf of a principal. In doing so, it reduces the need to rely on apparent authority doctrines and diminishes the risk of misattribution of actions.

A recent controversy involving Air Canada illustrates how these principles might play out in practice (Civil Resolution Tribunal (British Columbia), 2024). In this instance, the airline argued that it could not be held liable for information provided by its online chatbot. Implicitly, this suggests treating the chatbot as if it were separate from the airline—akin to an independent entity. Yet, in the judge's view, the chatbot exists as part of Air Canada's digital infrastructure and so the company was responsible for the information it provided. Under conventional principles of law and equity, the chatbot's outputs, even if generated autonomously, form part of the information the airline holds out to the public. The airline's attempt to evade responsibility runs counter to the principle that a firm must stand behind the representations it makes, whether through humans or machines. This case underscores that companies may be liable for the actions of their AI agents, a view also held by many scholars (Adler et al., 2024). From a broader perspective, this case also highlights the growing need for robust technological and legal mechanisms—like the authenticated delegation framework—that can delineate responsibility and authority in AI-mediated interactions, ultimately protecting consumer

trust and market stability.

Beyond agency law, existing legal frameworks for electronic transactions, like the Uniform Electronic Transactions Act (UETA), provide some guidance. The UETA is a uniform law adopted by 49 U.S. states to help accommodate the realities of e-commerce by recognizing that electronic communications and automated processes can play substantive roles in forming and executing agreements (Greenwood, 2024; National Conference of Commissioners on Uniform State Laws). Under UETA, parties are encouraged to adopt agreed-upon security procedures and error-detection protocols to ensure that the electronic records genuinely reflect the intended agreements. If one party fails to follow these procedures and an error that would have been detected goes unnoticed, the other party may be permitted to avoid the consequences of that error. Similarly, if an individual errs while interacting with an electronic agent and the system offers no reasonable correction mechanism, UETA contemplates relief for that individual under defined conditions.

These provisions reflect an understanding that trust in digital commerce requires more than just a willingness to be bound by electronic contracts; it also demands reliable methods for verifying authority, correcting mistakes, and ensuring that automated processes faithfully implement the intended instructions of the principal. The authenticated delegation framework aligns well with these goals. Integrating a verifiable chain of authority into interactions with AI agents provides the digital equivalent of an agreed-upon security procedure. In doing so, it can reduce misunderstandings and disputes about whether an AI-driven process was acting within the scope of its authority.

A critical element of both trust and accountability in AI-augmented systems lies in maintaining meaningful human oversight, often termed the "human-in-the-loop" requirement. The EU AI Act, for example, emphasizes the importance of maintaining human involvement in high-risk AI decisions to ensure ethical, transparent, and accountable outcomes (European Commission, 2021). The authenticated delegation framework supports this principle by making the human role in agent workflows explicit. Rather than delegating authority to an AI system behind opaque layers of code, third parties can firmly establish when, how, and under what conditions the AI is authorized to act. This allows humans to step in to verify decisions, correct errors, and ensure that automated actions remain aligned with overarching legal and ethical standards.

Strengthening the legal underpinnings, adopting frameworks for authenticated delegation, and integrating human oversight at critical junctures are all steps toward ensuring that emerging AI systems enhance market efficiency and maintain core values of trust, fairness, and accountability. Further empirical and doctrinal analysis could deepen this conversa-

tion, drawing on works that examine the real-world implementation of human-in-the-loop mechanisms (Mosqueira-Rey et al., 2023).

## 6. Alternative Views

A range of alternative views exist regarding the necessity and practicality of authenticated delegation for AI agents. One argument is that its complexity introduces unnecessary friction, potentially stifling innovation and discouraging adoption in rapidly evolving AI ecosystems. Another argument is that relying on OAuth 2.0 and OpenID Connect could further entrench large identity providers, raising surveillance and data monopolization issues. Additionally, the reliability of translating natural language permissions into structured access controls is questionable, as errors in interpretation or adversarial manipulation could undermine security rather than enhance it. There are also concerns about scalability, as implementing robust delegation workflows across diverse organizations, AI vendors, and dynamic use cases may prove impractical, leading smaller developers or institutions to favor more lightweight, ad hoc solutions. These alternative views suggest that while authenticated delegation provides a structured approach to AI governance, alternative models–such as decentralized identity frameworks, more flexible permission systems, or human-in-the-loop oversight without rigid delegation tokens–may better balance security, usability, and autonomy in AI-agent interactions. That said, we argue that perfect should not be the enemy of good: authenticated delegation has several real-world benefits that would immediately improve the reliability and trustworthiness of agentic systems. Introducing authenticated delegation systems, even if imperfect, would provide concrete security benefits, start the conversation on the optimal way to manage permissions, and lay the groundwork for further refinements.

## 7. Conclusion

This position paper argues the immediate need for authenticated delegation to AI agents, addressing urgent challenges around authorization, accountability, identity verification, and access control management in digital spaces. By extending existing OAuth 2.0 and OpenID Connect protocols with AI-specific credentials and delegation mechanisms, we propose a framework that enables secure delegation of authority from users to AI agents while maintaining clear chains of accountability. The proposed token-based framework—comprising user ID tokens, agent-ID tokens, and delegation tokens—provides a robust foundation for verifying agent identities, controlling permissions, and maintaining audit trails while supporting granular and robust scope limitations generated in response to natural language instructions. Our argument is supported by a detailed discussion of how

established internet-scale authentication (e.g., OpenID Connect and W3C VCs) and access management protocols (e.g., XACML) can be adapted to address the unique challenges of AI agent delegation while preserving compatibility with current systems, as illustrated through real-world use cases in areas like automated negotiations and web service interactions. As AI agents become more prevalent in digital spaces, frameworks like this will be essential for ensuring they operate within appropriate bounds while remaining accountable to their human principles. Looking ahead, key research directions include developing standardized scope definitions for common AI agent tasks, exploring privacy-preserving delegation mechanisms, and creating tools to help service providers implement and manage agent authentication policies, ultimately working toward ensuring AI systems can be safely and productively integrated into existing digital infrastructure.

## Impact Statement

The rapid emergence of AI agents presents a critical moment for designing accountability and safety into agent infrastructure. As these agents increasingly handle critical tasks and interact with a wide range of systems and stakeholders, robust mechanisms are needed to validate their identities, verify their permissions, and trace their actions become paramount.

## Acknowledgements

We thank Kim Hamilton Duffy, Ankur Banerjee, Steve McCown, Shrey Jain, and Raina Wu for their feedback and support on this work. Samuele Marro is supported by the EPSRC Centre for Doctoral Training in Autonomous Intelligent Machines and Systems grant n° EP/Y035070 and Microsoft Ltd.

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

# A. Extended Background

## A.1. Comparisons to other AI identifiers

To *verify human identity online*, a large body of work exists ranging from simple authentication such as OAuth 2.0 (Hardt, 2012) to more complex digital identity frameworks as W3C's Verifiable Credentials (Sporny et al., 2024b), decentralized identifiers (Sporny et al., 2024a), and the European Union Digital Identity's privacy-preserving digital wallets (Wallet, 2024). To *privately prove personhood*, a number of systems have been developed to distinguish human users from bots, including proof-of-personhood systems designed to counter automated Sybil attacks (Borge et al., 2017), simple turing tests such as CAPTCHAs (Von Ahn et al., 2003), and more robust credentials (Adler et al., 2024). More generally, the goal of 'know-your-customer' for users and granular access permissions (identity and access management, IAM) are commonplace on the internet.

Similarly, many websites seek to *broadly limit access to bots on their services*, and may do so through the use of robots.txt bans. This is important since the widespread presence of bots or unauthenticated AI agents can lead to abuse and harm, but is often done at the 'user-agent' level (for example, banning all 'GPTBot' user agents (Longpre et al., 2024)).

To *track and verify the output of AI systems*, watermarking techniques (Liu et al., 2024; Wang et al., 2021) and content provenance measures (C2PA, 2023) have emerged as potential solutions for determining the origin of AI-generated content. However, these approaches face reliability challenges (Saberi et al., 2024) and are insufficient for establishing comprehensive accountability or safety when using AI agents. The inherent limitations of current verification methods highlight the need for more robust frameworks that can track not just content creation but also the broader implications of AI system deployment and interaction.

For *managing access to sensitive AI capabilities* themselves, researchers have proposed 'know-your-customer' schemes for compute providers (Egan & Heim, 2023; O'Brien et al., 2023), while commercial platforms implement API tokens and access controls (OpenAI, 2023). These developments reflect a growing recognition that AI systems need robust mechanisms to prove their authenticity and permissions when accessing external services (Buterin, 2023), particularly as they become more integrated into critical infrastructure and decision-making processes.

To *identify specific instances of AI agents*, recent work has proposed identifiers and verification approaches discussed above (Chan et al., 2024b;a). This is important and critical work, which we build upon to extend to *authenticated delegation of AI agents* using existing authentication and permission protocols to enable AI agents to act on behalf of users in a controlled manner. In turn, these identifiers and delegation mechanisms can help create spaces that do not just gatekeep to human users but also enable AI agents to act on behalf of users with auditability and accountability.

## A.2. Documentation, safety, and governance of agentic AI systems

Documenting AI systems and the data that create them has been a critical area of research and practice. Early frameworks established foundational approaches including datasheets (Gebru et al., 2021), model cards (Mitchell et al., 2019), and data statements (Bender & Friedman, 2018), with popular implementations emerging (Paullada et al., 2021). Although each of these approaches has proven valuable, they face challenges in adequately addressing concerns around bias (Buolamwini & Gebru, 2018), privacy, and copyright. Recent work has highlighted the need for documentation of AI agents to understand their capabilities and limitations (Chan et al., 2024b), moving beyond static system descriptions to capture dynamic behaviors and interaction patterns. As AI systems become increasingly agentic, new frameworks are needed to document their evolving capabilities, decision-making processes, and potential risks (Bommasani et al., 2022).

## A.3. Governance of agentic AI systems

Recent work has explored runtimes for validating and reversing agent actions (Patil et al., 2024) and protocols for structured communication between language models (Marro, 2024). Researchers are also evaluating frontier models specifically for capabilities that could enable deceptive behavior (Phuong et al., 2024; Fang et al., 2024), while others advocate for tracking prior incidents (Wei & Heim, 2024) and establishing broader safeguards for AI agent interactions. Governance of AI agents is a rapidly evolving area of research and practice (Reuel et al., 2024; Kolt, 2024), with increasing attention being paid to the development of frameworks that can ensure responsible deployment and operation of autonomous systems.

# B. Technical Details

## B.1. Token-based authentication framework

Extending the existing OIDC framework, we can provide all relevant AI agent attributes and metadata of delegation in a set of identity-related tokens.

- *User's ID-token*: This is the existing ID-token data structure that is issued/signed by the OpenID Provider (OP) service. It is intended to represent information regarding the human user, and is no different to those

used in everyday login experiences.

- *Agent-ID token*: This carries the relevant information about that AI agent issued as an OAuth2.0 Native Client (meaning the owner of the AI Agent controls all keying material and secret parameters) and allows the corresponding service to verify any claims about the AI agent and its information. This token can include a range of additional information, from a unique identifier for the agent to a richer and more detailed agent ID token containing system documentation, capabilities or limitation metadata, relationship attributes to other AI systems, or other system characteristics. See Chan et al. (2024b) for further discussion of what an agent ID could entail.

- *Delegation Token*: This newly introduced token explicitly authorizes an AI agent to act on the user's behalf. The delegation token is issued and signed by the human delegator and carries references to (e.g., hash of) the corresponding user's ID token and the agent's Agent-ID token, allowing it to be verified by any service that trusts the OP. Further, any relevant information about the nature of the delegation can be shared. For example, sharing the summarized goal of the agent and its scope limitations could assist a third party in guiding the AI agent to useful endpoints and interaction paradigms. The delegation token should specify validity conditions, such as expiration time or revocation endpoints, and be digitally signed by the user to prevent forgeries and ensure that the user knowingly granted the AI agent the listed privileges. In addition, the token may carry supplemental metadata—for example, logging or audit URLs—allowing service providers to record interactions, monitor delegated actions, and respond appropriately to anomalies. By verifying that the delegation token references a valid user ID-token and a properly issued agent ID-token, remote services can confirm the authenticity and scope of the AI agent's authority before granting access.

## B.2. Using verifiable credentials as an alternative

The W3C Verifiable Credentials (VC) standard (Sporny et al., 2022) offers a versatile alternative—and sometimes complement—to existing OpenID Connect (OIDC) flows for conveying identity and delegation data. Under a VC-based approach, an issuer (such as an organization or individual) can sign a credential that attests to various claims about a subject, which might be a user, an AI agent, or any other entity needing verifiable, tamper-evident attributes. Because VCs are not bound to a particular transport protocol, they can be presented and verified in a decentralized or peer-to-peer manner without always relying on a single identity provider. This contrasts with OIDC, which gener-

ally depends on a central OpenID Provider (OP) to mint and validate tokens.

A key benefit of VCs is their privacy-enhancing potential. Rather than disclosing all attributes or relying on a single identity provider, users, and AI agents can share only the subset of claims strictly necessary for a given interaction. This "selective disclosure" capability can mitigate concerns around centralized logging or cross-platform correlation inherent in OIDC-based architectures, especially when interactions span multiple domains or organizations.

Nonetheless, replacing OIDC entirely with a purely VC-based model does come with trade-offs. OIDC already enjoys a robust ecosystem of libraries and deployments that provide well-tested support for issues like token refresh, revocation, and audience restriction. VCs, while powerful, require additional work to replicate these flows at scale—particularly if each verification call demands a new signature check or interaction with a blockchain or distributed ledger. In many enterprise environments, stakeholders may prefer to incorporate VCs into existing SSO or multi-factor authentication frameworks, rather than adopt a fully decentralized identity infrastructure upfront.

In practice, hybrid solutions often prove the most pragmatic. A user or AI agent could store and manage VCs encoding rich attributes or regulatory endorsements, while still leveraging OIDC tokens to bootstrap compatibility with existing authentication or authorization endpoints. For instance, an Agent-ID token could embed a VC carrying detailed metadata on its behavioral, property, context, and relationship attributes. Service providers integrating with OIDC get the familiar token-based handshake, while still retaining the option to parse the embedded VC for an additional layer of trust and context. Examples such as OID4VC support this (Yasuda et al., 2022).

## B.3. Federated OpenID Providers for Agent Mutual Authentication

One of the key goals of AI Agents is the ability for an agent to interact with existing web services as well as other AI Agents (AI Systems). To enable secure interactions, AI Agents must perform mutual authentication and verify each others' credentials, including Agent-ID tokens and delegation tokens.

The authentication flow begins when agent A1 presents its Verifiable Credential to agent A2. The VC contains claims that must be validated through the respective OpenID Provider, including the user's ID-token and the Agent-ID token. Since the APIs at OP1 are protected, A2 must authenticate itself using its own Agent-ID token previously issued by OP2 in its home domain. This cross-domain verification is achieved through federation, where OP1 validates

A2's credentials by communicating with OP2. While the figure demonstrates authentication from A1's perspective, the process is mutual, ensuring both agents can verify each other's delegated authorities and credentials through their respective OpenID Providers.

## B.4. Identification of AI Systems and AI Agents

One of the challenges facing the deployment of AI technologies is the need to establish identification mechanisms for instances of AI systems, including AI Agents (Chan et al., 2024b). Here it is useful to distinguish two basic types of identifiers:

- *Local identifiers*: A local identifier is a unique string (e.g. UUIDv2) that can be used to distinguish an instance of an AI system from another within a given domain. This means that other systems and entities in the domain are able to pinpoint each AI system using that local identifier. A local identifier may be meaningless outside the domain, and thus require a mapping to a global identifier.

- *Global Identifiers*: A global identifier enables an AI system to be referred to (or referenced to) from anywhere in the Internet. This enables agents to interact with other AI systems and other AI agents across different geographies.

  From a scalability perspective, it is useful to be able to map from the global identifier of an AI agent to its local identifier to enable other systems within its home domain to provide support for that AI agent, such as a local authentication by the OP in that home domain that attest to the true existence of the agent within the domain.

  A global identifier belonging to an AI system or agent can be incorporated within a *decentralized identifier* (DID) structures (W3C, 2021) that then enables useful interactions with DLT based services that function based on the DID.

Due to the prevalence of OAuth 2.0 and OIDC deployments today, it is useful to reuse some of the existing identifier structures already utilized in these deployments. If we view an AI Agent as being a client (native or hosted service) within OAuth 2.0 then we could reuse the two important parameters used by an OAuth 2.0 client to interact with the authorization server (or the OP). These parameters are the `client_id` and the `client_secret` (see section 2.3.1 of (Hardt, 2012)). The client-ID and the client-secret parameters in OAuth 2.0 is used by the authorization server (the OP) to recognize a client that had been previously registered to the OP using specific client registration protocols (Jones

et al., 2015). In the current context of identifying AI systems and AI agents, the client-ID could be considered a local identifier that is meaningful only in the domain serviced by the specific OP (i.e. where the client has been registered). However, the client-ID could be the basis for the OP to issue a *delegation token* that signifies the user delegating authorization to their AI Agent to carry out certain tasks, defined by *action scopes* within the delegation token.

## B.5. ID token threat model

Our proposal is meant to be secure against several different security threats, ranging from the authenticity of the issued tokens to the nature of the delegation.

With respect to ID tokens, Chan et al. (2024b) identifies three fundamental threats that an AI ID system must defend against. The first threat is tampering, where an attacker modifies the ID while it is being transmitted between the author and the receiving party, potentially altering crucial system information or attributes. The second threat is ID spoofing, where an attacker creates a fraudulent ID and falsely claims it originated from a legitimate author (such as a major AI company), which could enable malicious systems to masquerade as trusted ones. The third threat is instance spoofing, where an attacker takes a legitimate ID and attempts to use it with their own unauthorized AI instances, essentially hijacking the reputation or privileges associated with the original system. To counter these threats, the authors propose that IDs must implement digital signatures that cover both the ID itself and the system's outputs, similar to how HTTPS certificates work for websites. However, they note an important limitation: since the signature must cover both ID and output, any modification to the output (even benign ones) would invalidate the ID, creating a challenging trade-off between security and usability. These threats to robust AI system identification extend naturally to the task of authenticated delegation for AI agents, which requires robustness for both AI system verification, human delegate verification, and verification of valid delegation.

OpenID Connect could help prevent several additional threats beyond these robust IDs. Through its built-in mechanisms, OIDC could prevent identity correlation attacks by using pairwise pseudonymous identifiers to ensure AI instances appear different to different services, thwarting attempts to track instance behavior across platforms. Its session management capabilities could prevent session hijacking attempts against active AI instances, while its dynamic client registration could prevent impersonation through unauthorized endpoints. Most significantly, OIDC's scoping and audience restriction mechanisms could prevent authorization scope abuse and cross-instance privilege escalation, ensuring AI instances cannot exceed their intended permissions or use tokens meant for other instances. The protocol's

discovery mechanisms could also prevent identity provider spoofing, adding another layer of security to the ID ecosystem.

# C. Extended discussion of scoping for AI agents

We distinguish between **task scoping** and **resource scoping**:

- Task scoping involves specifying which actions or workflows an agent is authorized to perform on behalf of the user. These actions may range from high-level tasks (e.g., "draft a financial report") to more granular actions (e.g., "create a new database entry");
- Resource scoping involves specifying which resources (information, APIs, tools, etc.) an agent can use or modify.

While conceptually distinct, task scoping and resource scoping are closely connected. Limiting which tasks can be performed also means that a (well-designed) agent will not access unnecessary resources; similarly, restricting access to specific resources also constrains what tasks are feasible in the first place.

## C.1. Structured permission languages

A large class of scoping mechanisms relies on structured, machine-readable policy specifications. These specifications unambiguously define which entities have which authorizations, under which conditions, and with what privileges. Several well-known languages and frameworks exist for encoding permissions, such as XACML (eXtensible Access Control Markup Language), which uses XML to encode and evaluate access control policies (OASIS, 2013), and ODRL (Open Digital Rights Language), designed for expressing usage permissions over digital content (W3C, 2018). Other languages include OBAC (Brewster et al., 2020), ROWL-BAC (Finin et al., 2008), KaOS (Van Lamsweerde, 2001) and Multi-OrBAC Abou El Kalam & Deswarte (2006), which rely on ontologies (typically described using OWL) to model resources, subjects, and authorizations. In web-based contexts, this can often be as simple as whitelisting or blacklisting URLs and subdomains that an agent can access.

These structured languages are machine-readable and can thus be enforced reliably by traditional (non-AI) systems. From a practical perspective, they are well-suited for resource scoping, since resources are typically discrete and can be classified, enumerated, and grouped into security domains. For instance, when a policy states that a certain directory is read-only for a particular agent, enforcing compliance is straightforward and can be implemented at the system level.

However, they have three main drawbacks. First, while these frameworks are suitable for enumerating resources, they are less flexible for task scoping, especially when tasks are open-ended or cannot be easily described as a set of operations. Second, policy definitions can become lengthy and complex, especially in environments with a large number of resources and tasks, or in web contexts where the number of possible web interactions is enormous. Third, they are often environment-specific and require updating for different digital systems with which an agent interacts.

Despite these drawbacks, structured permission languages remain a cornerstone of access control because they provide a precise, easily auditable basis for resource scoping. An alternative approach involves using *schema validation* to constrain how agents interact with the environment, discussed in Subsection C.2

## C.2. Schema Validation As Scoping

An alternative approach to structured permission languages involves using *schema validation* to constrain how agents interact with the environment. In this approach, an AI agent's possible outputs or queries must conform to a predefined schema (Allemang & Sequeda, 2024). For example, if the agent can only communicate using RDF tuples, the system can enforce rules on the permissible classes, properties, or relationships that the agent can generate.

In practice, schema validation can be particularly powerful in scenarios where the system is designed around standardized data formats (e.g., JSON, XML, RDF). By restricting the agent to these formats and validating every output (e.g. using JSON-Schema (ECMA, 2017) or SHACL (W3C, 2017)), schema validation indirectly controls which actions are feasible. For instance, if an agent is only allowed to generate RDF triples with certain predicates (e.g., "hasTitle" or "hasSummary") and certain classes (e.g., "Document"), it cannot arbitrarily mutate data outside of that schema domain.

Like structured permission languages, non-AI systems can quickly and deterministically verify if an agent's output complies with a given schema. Moreover, since the output of the agent is already structured, schema validation may be simpler compared to parsing unstructured text. Standard outputs also simplify logging, as every action can be captured and audited with structured queries.

On the other hand, a rigid schema can reduce flexibility, especially in the context of task scoping. Tasks that require nuanced or creative outputs can be difficult to capture in a schema-based approach without introducing significant complexity (especially if such tasks evolve over time). Moreover, designing a robust schema that is both expressive and safe requires considerable effort, and the agent must be trained or prompted to work exclusively within that schema.

Nevertheless, schema validation can be a powerful mechanism for resource scoping, particularly when the range of permissible actions can be codified in a structured format.

## C.3. Authentication flows

Another dimension of controlling agent behavior is the **authentication flow** (i.e., deciding when to prompt a user or another authority for confirmation before the agent proceeds with an action). Rather than frontloading all access decisions into a single policy definition, an authentication flow can dynamically request user approval for borderline or high-risk operations.

The main advantage of this approach is that users do not need to define every edge case in a static policy. Additionally, authentication flows can be combined with other scoping mechanisms: for example, a policy can state that any resource that is neither explicitly approved nor explicitly forbidden requires human approval.

On the other hand, frequent authorization prompts can negatively affect the user experience, leading to "prompt fatigue" (Baruwal Chhetri et al., 2024), where the user simply grants permissions without a proper review. Moreover, determining when a request requires explicit authorization can be non-trivial, and misclassifications can lead to either excessive prompting or critical operations slipping through unnoticed.

In practice, a well-designed system can combine robust, structured policy definitions (for common scenarios) with dynamic authentication flows for rare or particularly sensitive actions. This approach allows users to offload the majority of routine checks to automated policies while still preserving the ability to escalate novel or ambiguous requests for user confirmation.

## C.4. Natural Language Mechanisms

Alongside fine-tuning, prompting has often been employed to steer the behavior of a model towards safety (Zheng et al., 2024). A reasonable extension of this approach would be to train (or prompt) the LLM to interpret permissions described in plain language. For instance, a user might say, "You are allowed to generate summaries of public documents, but you must not reveal any confidential metrics." Such instructions can, in principle, be parsed and acted upon by an LLM-based system.

The main strength of this paradigm is its user-friendliness. Non-technical users may find expressing policies in natural language much easier than writing formal rules. Moreover, natural language can capture nuanced or context-dependent instructions that are difficult to encode in structured languages. This makes them ideal for both task and resource scoping. Finally, natural language can be used to enforce policies on actions that require reading or using natural language, such as interactions with other LLM-based agents.

However, natural language often lacks the precision needed for reliable policy enforcement. For instance, terms like "sensitive data" or "private emails" may be interpreted differently depending on context. This problem is particularly relevant in the case of conflict between different policies, where ambiguous and context-dependent instructions may yield different interpretations. Relying solely on an LLM to interpret and enforce ambiguous natural language instructions can be risky in security-sensitive contexts.

In short, while natural language instructions can serve as a convenient mechanism (especially for task scoping, where other mechanisms are less suitable), they are **not** reliable enough to be used as standalone policy mechanisms.

## C.5. Controlled Natural Languages

While natural language permissions are flexible, they lack specificity. Controlled Natural Languages (CNLs) (i.e., subsets of natural language with restricted grammar and vocabulary), represent an interesting middle ground between structured and freeform specification. They preserve some of the readability of natural language while being more suitable for automated parsing and formal verification. An agent using a CNL interface might be able to interpret requests unambiguously, which reduces the risk of accidental misinterpretation. However, designing a CNL that is both secure and expressive can be challenging: allowing too much freedom increases ambiguity and exposes LLMs to prompt injection attacks (Perez & Ribeiro, 2022), while a CNL that is too restricted will suffer from the same issues as structured languages.

## C.6. How this interacts with robots.txt

Robots.txt has, without legal heft, underpinned the modern web for decades. It relies upon a simple set of directives, where a user-agent is given rules for a subroute. Just as the recent proliferation of scraping has led to rapid uptake of new user-agent rules Longpre et al. (2024), new directives could easily be rolled out across the web with the right incentives.

This system still has a place in a web full of AI agents. While websites may wish to block scraping, they may also wish to guide agents to the correct subroutes where they could share credentials and interact. For example, a website may wish to block scraping, allow human users to interact, and send AI agents directly to an API natural language interface designed for AI systems.

To guide agents to the correct subroutes where they could share credentials and interact, we can define a new user agent, `AgentBot`, and force it into a specific interaction

route (e.g., `/AgentInterface/`). Since `robots.txt` is a guide, not a rule, this route can go on to provide richer details of what services can be accessed and what sitemaps exist. Such a `robots.txt` need only be an initial guide to agents.

## C.7. Inter-agent scoping.

Extending beyond the user-agent-service model, this approach can apply to multi-agent settings where agents want to propagate their limitations onto other agents performing actions on their behalf. Suppose that the user specifies the authorizations of an agent Alice. When Alice interacts with another agent, Bob, in natural language to perform a task, Bob can parse Alice's scoping instructions and interpret them in its own environment. By doing so, Bob can confirm that its assigned operations remain within the original scope, and provide an auditable receipt of the actions taken and the resources accessed. This is particularly useful in scenarios where inter-agent communication spans different organizations, each with separate policies and resource constraints.

For a concrete example, suppose that Alice is a project management agent and Bob is an accounting agent. The user describes in plain English a financial data request to Alice; Alice thus sends the forwarded request and a description of the authorizations to Bob. Bob replies with a structured interpretation of the authorizations (e.g., "Read-only access to 'transactions2025' dataset, columns: total amount, vendor name"), which is logged and approved by either the user or Alice.

Such a workflow ensures that even if the agents communicate in flexible natural language, their underlying scoping and record-keeping remain anchored in auditable, deterministic policy. As a result, the risk of unauthorized data sharing or unbounded agent behavior is greatly reduced, and each agent's capacity to "inherit" restricted credentials from the delegator is tightly controlled.

# D. Limitations of the current proposal

## D.1. Problems with an OpenID Connect approach

While the OpenID Connect (OIDC) and OAuth 2.0-based framework proposed here provide robust and battle-tested mechanisms for authentication and delegation, it comes with trade-offs and may be more complex than alternatives with different trade-offs in privacy, security, and auditability.

**Overhead from multiple sign-in flows.** A significant drawback of the OpenID Connect approach is the potential overhead introduced by multiple sign-in flows required to authorize AI agents across individual service providers. This can be likened to the experience of setting up a new

email client, where users must repeatedly log in to authorize access to various services. While such authorization flows enhance security by ensuring each provider independently verifies the AI agent's delegation credentials, they impose a usability cost by slowing down access to secure systems. In theory, it is possible to bypass this burden by presenting delegation tokens directly without performing the full OIDC authentication flow; however, this shortcut sacrifices key security guarantees, particularly those related to token freshness and verification.

**Increased reliance on OpenID Providers and privacy risks.** The reliance on OpenID Providers (e.g., Google, Facebook, or equivalent entities) introduces systemic privacy concerns. Since OIDC providers mediate all authentication flows, they gain the ability to track and correlate individual AI agent interactions across various services. This can include collecting statistical usage analytics or requiring relying parties to share logs, which facilitates extensive behavioral profiling. Such centralized visibility undermines user privacy and creates a potential single point of surveillance. Addressing these risks necessitates strong privacy mitigations, such as pairwise pseudonymous identifiers or the minimization of log-sharing requirements, but these mechanisms add further complexity to the system.

**Comparative complexity relative to W3C Verifiable Credentials.** While the paper highlights the ability to embed W3C Verifiable Credentials (VC) within the OIDC framework, the full OIDC authorization flow may still be unnecessarily heavy compared to native W3C VC-based delegation and authentication processes. W3C VC issuance, authentication, and delegation mechanisms could directly fulfill the same requirements for AI agent identity verification without incurring the additional overhead of repeated authorization flows and central provider mediation. Additionally, W3C VC-based approaches are inherently more privacy-preserving, as they do not rely on a single provider to mediate trust or track credential usage. A streamlined VC-based process could generate OIDC-compatible credentials when required, enabling interoperability while preserving simplicity and privacy. Similarly, other proposed alternatives to OAuth 2.0 specifications could be drop-in solutions here to address design trade-offs, such as the Grant Negotiation and Authorization Protocol (GNAP) (Richer & Imbault, 2024). Further exploration of these alternative approaches remains essential to determine their feasibility as lightweight solutions for AI agent delegation.

Taken together, these limitations highlight key trade-offs between security, usability, and privacy in the OIDC-based framework. While the proposed approach remains an incremental and interoperable path forward, addressing these challenges will be critical to ensuring a robust and practical

system for AI agent authentication and delegation.

## D.2. Limitations of natural language scoping

Although translating natural language scoping instructions into structured permission languages enables a more flexible interface, it also creates several key challenges.

**Evaluating reliability and correctness.** One of the foremost difficulties is ensuring that the translation from a user's natural language specification to a machine-readable policy is accurate. Natural language instructions often contain context-dependent or ambiguous terms, making them inherently prone to misinterpretation by an AI system. Although a human-in-the-loop approach can mitigate these risks through policy review, such human verification is not infallible; users may inadvertently miss subtle translation errors. Moreover, as the complexity of a permission specification grows, verifying the alignment between the original natural language instruction and the generated structured policy becomes more difficult, both technically (due to large policy definitions) and cognitively (due to the burden on human reviewers).

**New threat vectors for LLM attacks.** Exploiting weaknesses in language-based interfaces can expose novel threats that do not exist under purely static access control. Prompt injection and jailbreak attacks can coerce a large language model into generating or accepting policies that exceed the original users intent, thereby gaining unauthorized privileges. While separating resource or task-scoping instructions from normal chat sessions or interactions reduces the likelihood of an attack, it still presents a new differentiated attack surface that needs to be guarded.

**Contextual drift.** As policies evolve or the task context changes over time, prior natural language instructions risk becoming outdated or misaligned with newly introduced resources. Maintaining consistency across multiple revisions of instructions is nontrivial.

**Partial reliance on third parties to enforce restrictions.** In some contexts, the access control rules are applied to an external environment or agent that is being interacted with. To maintain security over the application of these access controls, it may be necessary for the corresponding party to enforce the rules beyond trusting the native agent to follow them. In such instances, the reliability of the third-party becomes a critical point of failure.

## E. Example Use Cases

This section outlines four scenarios where authenticated delegation ensures secure and accountable AI agent interac-

tions. Each example illustrates the structure of delegation credentials, the scoping mechanisms they enforce, and their role in maintaining accountability.

### E.1. AI Agent for Web Browsing

**Scenario.** A user employs an AI agent to perform tasks such as scheduling appointments, retrieving information, and managing online payments. The agent's access must be restricted to specific websites, with clear limitations on the actions it can perform, such as transaction amounts.

**Approach.**

1. **Delegation Credential.** The credential specifies:

   - *User Identity:* The unique identity of the delegating user.
   - *Agent Identity:* A unique identifier for the agent, including its capabilities (e.g., browser-based interactions).
   - *Scope:* Restrictions such as approved websites, permitted actions (e.g., viewing schedules, making payments), and specific constraints (e.g., spending limits, validity duration).

2. **Access Enforcement.** Websites validate the agent's credential upon login or transaction attempts. Unauthorized actions, such as accessing unapproved sites or exceeding predefined limits, are automatically blocked.

3. **Auditability.** Logs tied to the agent's unique identity record all transactions and actions, enabling post-interaction review and traceability.

**Why It Matters.** The structured credential ensures the agent cannot access unauthorized websites or perform unintended actions. This protects sensitive user data and ensures the user retains control over their online interactions.

### E.2. API-Only Data Manager

**Scenario.** An organization uses an AI agent to aggregate and analyze data from internal APIs, such as those providing information about operations or inventory. The agent's access must be restricted to specific APIs and limited to non-destructive actions like querying data.

**Approach.**

1. **Delegation Credential.**

   - *User Identity:* The authenticated identity of the delegating organization or individual.
   - *Agent Identity:* A unique identifier for the agent, specifying its purpose (e.g., data aggregation).

- *Scope:* Access permissions restricted to specific APIs, with limitations on actions (e.g., read-only access) and operational constraints (e.g., rate limits or expiration).

2. **API Enforcement.** APIs validate the credential and deny actions outside the granted permissions, such as attempts to write data or access restricted endpoints.

3. **Credential Management.** Delegation tokens are periodically rotated or updated to reduce risks associated with stale credentials.

**Why It Matters.**    The agent's restricted scope ensures it cannot alter or access sensitive data unintentionally. Detailed access logs provide accountability and enable quick responses to anomalous behavior.

### E.3. Remote Virtual Environment via SSH

**Scenario.**    A user directs an AI agent to execute tasks in a remote virtual environment, such as running simulations or processing data. The agent's access must be limited to specific commands and directories.

**Approach.**

1. **Delegation Credential.**

   - *User Identity:* The user's authenticated identity with the virtual environment provider.
   - *Agent Identity:* A credential tied to the agent, specifying its role (e.g., simulation execution).
   - *Scope:* Permission to access specific directories, execute defined commands, and perform actions within a restricted time frame.

2. **Environment Enforcement.** The server enforces access control policies. Unauthorized actions, such as modifying configuration files or accessing sensitive directories, are rejected.

3. **Audit Trail.** The environment logs each command executed by the agent, tied to its unique delegation credential, for post-task review.

**Why It Matters.**    The restricted delegation credential ensures that the agent operates only within its assigned scope, safeguarding the environment against unintended or malicious actions.

### E.4. Agent-to-Agent Collaboration

**Scenario.**    Two AI agents collaborate on a complex task, such as event planning or contract negotiation. Each agent has distinct roles and permissions that must be respected, such as one handling logistics and the other managing finances.

**Approach.**

1. **Delegation Credentials.**

   - *User Identity:* The authenticated identity of the delegating organization or individual.
   - *Agent Identities:* Each agent receives a unique credential describing its role and capabilities.
   - *Scopes:*
     - Agent 1: Permissions for logistical tasks, such as booking services or scheduling.
     - Agent 2: Permissions for financial tasks, such as processing payments, with explicit budget constraints.
   - *Cross-Agent Verification:* Each agent includes its credential when issuing requests to the other. The receiving agent verifies the request is within scope before proceeding.

2. **Collaboration Mechanism.** The agents communicate using natural language, but all actionable requests reference their credentials for validation.

3. **Auditability.** A log of all interactions, including credential references, ensures a clear record of tasks and decisions.

**Why It Matters.**    By embedding scoping rules into cross-agent interactions, the collaboration remains secure and accountable. Each agent operates within its predefined limits, reducing the risk of unintended actions or miscommunications.

