# OpenReview forum: "Position: AI Agents Need Authenticated Delegation"
_ICML.cc/2025/Position_Paper_Track — ICML 2025 Position Paper Track oral_

### Official Review · Reviewer_4V3X · 2025-02-18

**Significance:** 4
**Argument Clarity:** 4
**Rating:** 4
**Confidence:** 3

**Questions:**

na

**Discussion Potential:**

3

**Paper Summary:**

This paper argues that authenticated and auditable delegation of authority to AI agents is crucial for mitigating risks and unlocking the value of AI systems. It proposes extending existing web authentication protocols like OAuth 2.0 and OpenID Connect to enable secure delegation of authority to AI agents. The authors suggest using structured permissions and natural language interfaces to translate user instructions into robust access control rules. The paper also explores how this framework can ensure accountability and compatibility with existing web infrastructure. It provides a roadmap for implementing authenticated delegation, including legal implications and example use cases.

## update after rebuttal
No update.

**Position:**

Yes

**Position In Title:**

Yes

**Related Work:**

3

**Strengths And Weaknesses:**

S1.The paper tackles the urgent challenge of ensuring accountability and security when AI agents interact with third-party services or other agents. This is particularly relevant given the rapid deployment of autonomous AI systems.

S2.By leveraging existing protocols like OAuth 2.0 and OpenID Connect, the proposed framework can be integrated into current web infrastructure without requiring a complete overhaul. This ensures immediate compatibility and practicality.

S3.The paper combines multiple strategies, including structured permissions, natural language processing, and human oversight, to create a robust delegation framework. This hybrid approach balances flexibility and security.

I think this paper effectively supports its argument and does not have apparent weaknesses.

**Support:**

4

---

> ### Author Rebuttal · Authors · 2025-03-25
>
> Thank you for the review and your comments! We hope this topic sparks some interesting conversation in the community, much of which we have already seen in other contexts. We hope to expand on this work moving forward to provide some robust solutions for the community.

---

### Official Review · Reviewer_b9Gt · 2025-03-01

**Significance:** 3
**Argument Clarity:** 4
**Rating:** 4
**Confidence:** 3

**Questions:**

In my opinion, the paper is clearly written, so I don’t have any clarification questions at the moment. Nevertheless, I would appreciate it if the authors could provide more details about the alternative views discussed in Section 6.

**Discussion Potential:**

4

**Paper Summary:**

The paper argues for the need for authentic and auditable delegation to AI agents. The paper first explains why this aspect is important for agentic AI, explaining the risks that it could mitigate. The paper then elaborates on how to enable authentic delegation for AI agents by extending existing protocols and describes a procedure for defining permissions for AI agents.

---
## update after rebuttal

I thank the authors for their response. I will keep my original positive score.

**Position:**

Yes

**Position In Title:**

Yes

**Related Work:**

3

**Strengths And Weaknesses:**

**Strengths**

- The paper is well-written, enjoyable to read, and easy to follow. It clearly states its position from the beginning, motivating it with a plethora of examples and connecting it to existing delegation protocols.
- The appendix provides an extensive overview of existing literature related to the topic, while the main paper focuses on the most important references supporting its argument.
- The arguments are clearly stated and supported with several examples that demonstrate the need for authentic delegation in AI systems.
- The paper describes a high-level extension of OAuth 2.0 and OpenID protocols that could be adopted to authorize AI agents to carry out tasks on behalf of users.
- The paper recognizes the need for resource scoping and outlines a method for defining permissions, which would then be translated into a permission language.
- The discussion in Section 5 is compelling and provides arguments grounded in legal frameworks that support the paper’s position.
- In my opinion, the position argued in the paper is intriguing. The topic is highly relevant and would likely lead to interesting discussions.

----

**Weaknesses**

- As a reader who is not fully familiar with the landscape of authentication protocols, I believe it would be valuable to provide a more in-depth overview of authentication protocols and compare them (e.g., in the appendix).
- Section 6 could be expanded. A more detailed discussion on alternative views could be provided. Adding references that support these alternative views would be valuable.
- The paper is well-written, but there are minor typos (particularly in the appendix).

---

**Support:**

4

---

> ### Author Rebuttal · Authors · 2025-03-25
>
> Thanks for your detailed review. The fact that you found typos in the appendix that we missed makes us appreciate your attention to detail (these will be fixed)!
>
> We agree with several of your points, and have found that this topic has already led to a range of interesting discussions. With respect to the weaknesses you highlight:
> 1) Authentication protocols landscape: this is indeed a giant field in its own right, with a rich history, literature, and many solutions. While many parts of this history are not relevant to the current AI challenges and the position, we agree that there is more to be said in summarising the alternatives. We have also independently heard this feedback from other parts of the community. We will update the manuscript (likely in the appendix due to space constraints) with additional references and commentary on this such that a reader can feel more comfortable with the full spectrum of authentication protocols.
> 2) Section 6, alternative views: Similar to above, there is more that could be added here, including alternative approaches (e.g., other authentication flows) and the perspective that none of this matters at all. Section 6 is important, and we can expand this to add references to these alternative views.
>
> Thanks again for your positive review.

---

### Official Review · Reviewer_mB52 · 2025-03-14

**Significance:** 4
**Argument Clarity:** 4
**Rating:** 4
**Confidence:** 2

**Questions:**

- Have you considered how your proposed extension of OAuth 2.0 and OpenID Connect would perform at scale?

**Discussion Potential:**

4

**Paper Summary:**

This paper highlights the need for secure, auditable authority delegation to AI agents to address risks related to their autonomous operations. It advocates extending OAuth 2.0 and OpenID Connect with AI-specific credentials and emphasizes transforming natural language permissions into structured access control policies. The paper provides technical recommendations and use cases demonstrating the security and clarity benefits of authenticated delegation, while also acknowledging practical limitations and alternative implementation views.

**Position:**

Yes

**Position In Title:**

Yes

**Related Work:**

4

**Strengths And Weaknesses:**

I am not an expert in this area and I try to outline some strengths and weaknesses from my point of view.

Strengths:
- The topic is timely, relevant and important. Safefy in AI agents is important and this paper addresses an urgent and contemporary challenge of authorization, accountability, and access control for AI agents.
- This paper is well motivated and novel. As far as I know, few works has been done on Agent's authenticated delegation, extended from OAuth2, but it seems important on the actual deployment of AI agents.
- This paper is well written. All arguments are well presented and supported.
- I appreciate the discussion of legal grounding for authenticated delegation.

Weaknesses:
- No obvious weaknesses found.

**Support:**

3

---

> ### Author Rebuttal · Authors · 2025-03-25
>
> Thank you for the review. We agree with all your comments and think you've captured the spirit of the paper well.
>
> The question you raised about the scalability of OAuth 2.0 and OpenID Connect is fantastic. These were chosen partly because of their wide use and experience in scaling to massive deployments. With respect to their ability to computionally be feasible at a massive scale, we have clear evidence this will work. However, these are important questions about how human involvement in authorisation and delegation control scales as AI systems become more autonomous. This is discussed in the paper but has also spurred even more discussion outside of it, which ultimately was the goal of this paper.
>
> Thanks again for your review.

---

### Official Review · Reviewer_Ut3L · 2025-03-17

**Significance:** 4
**Argument Clarity:** 4
**Rating:** 4
**Confidence:** 3

**Questions:**

N/A

**Discussion Potential:**

4

**Paper Summary:**

This position paper advocates for authenticated and auditable delegation in AI agents to ensure accountability and security. It first highlights risks (e.g., lack of transparency, prompt injection attacks, etc), and proposes extending OAuth 2.0 and OpenID Connect with AI-specific credentials to allow third parties to verify an agent’s identity and permissions. The authors also presents a technical framework for secure delegation, outlining how natural language permissions can be translated into structured access rules.

**Position:**

Yes

**Position In Title:**

Yes

**Related Work:**

4

**Strengths And Weaknesses:**

**Strengths**

(1) Compelling Support: The authors provide well-reasoned arguments, backed by literature and real-world use cases, illustrate the risks and feasibility of the proposed solution. For example, In Section 2, the examples are concrete and well motivated.

(2) Clear Position and concrete solution: the authors advocate proposes extending OAuth 2.0 and OpenID Connect with AI-specific credentials to allow third parties to verify an agent’s identity and permissions. The solution is very concrete and the author also states clearly on its limitations (challenges of translating natural language into structure format)

(3) Well-Organized and Cited: The paper is clearly structured and well written. For example, it provides clear terminology and extensive references across ML, security, and law.


**Weaknesses**

(1) A prototype demonstration of the proposed framework would enhance credibility of the position paper.  Currently, the paper is mainly conceptual, which is expected for a position paper). However, demonstrating a simple prototype – for example, an AI agent using an actual OAuth extension to perform a task with delegated credentials, can make address the feasibility concerns.

(2) More discussion on how ML models handle permission translation and delegation learning could strengthen relevance. For example, the discussion in the paper focuses on system design and security protocols. Expanding on how the delegation framework might integrate with the development of AI agents (e.g., implications for training agents to understand scoped permissions can make the work even more relevant to core ML research).

(3) Improve presentation on glossary. A glossary or table summarizing delegation components (e.g., user tokens, agent ID tokens) would improve readability, instead of just putting them into the appendix.

**Support:**

4

---

> ### Author Rebuttal · Authors · 2025-03-25
>
> Thank you for the thoughtful review and positive feedback regarding the structure and motivation.
>
> The weaknesses you outline are great points, with which we have much to add:
> 1) Prototype: We have several! The nature of the perspective piece was to spark conversation, which offline and in other forums it has. This has resulted in multiple prototypes, each of which takes a different approach to the problem. This can include structuring the auth management around protocols like MCP, focusing on multi-agent authentication, or determining the extent to which services should treat the identity provision as a general web service, a subtype of a human user, or a different category altogether. Many of these questions are 'in the weeds,' but we can add content to the final manuscript outlining some key insights from these different prototypes that have been created since January.
> 2) The current position focuses on system design, but the question of examining how to train models specifically with respect to permission translation and delegation learning is excellent. We don't believe this can fit in the current scope of this paper, but we will note it as future work.
> 3) This is a big, and evolving topic, and a glossary would go a long way. We will make edits to the paper for clarity.
>
> Thanks again for your review, and we hope this sparks positive discussion amongst the community.

---

> > ### Comment · Reviewer_Ut3L · 2025-04-03
> >
> > Thank you for your response, the rebuttal makes sense to me. I will keep my score.

---

### Decision · Program_Chairs · 2025-04-26

**Decision:**

Accept (oral)

**Comment:**

All reviewers were extremely positive about the timeliness, relevance, and interest of the position. The paper is well written and the argumentation very convincing. The paper proposes concrete ideas, while being realistic and honest about potential issues and limitations.
The replies to the minor weaknesses pointed out by the reviewers are convincing - some small changes to the paper and additions to the appendices will further strengthen the paper, some of the others indeed go beyond what is feasible within the scope of a position paper and the page limit, and fall into "additional ideas and discussions sparked by the paper" rather than weaknesses.